# The Magnetic Behaviour of CoTPP Supported on Coinage Metal Surfaces in the Presence of Small Molecules: A Molecular Cluster Study of the Surface *trans*-Effect

**DOI:** 10.3390/nano12020218

**Published:** 2022-01-10

**Authors:** Silvia Carlotto, Iulia Cojocariu, Vitaliy Feyer, Luca Floreano, Maurizio Casarin

**Affiliations:** 1Dipartimento di Scienze Chimiche, Università degli Studi di Padova, Via Francesco Marzolo 1, 35131 Padova, Italy; 2Istituto di Chimica della Materia Condensata e di Tecnologie per l’Energia (ICMATE), Consiglio Nazionale delle Ricerche (CNR) presso Dipartimento di Scienze Chimiche, Università degli Studi di Padova, Via F. Marzolo 1, 35131 Padova, Italy; 3Peter Grünberg Institute (PGI-6), Forschungszentrum Jülich GmbH, Leo-Brandt-Straße, 52428 Jülich, Germany; i.cojocariu@fz-juelich.de (I.C.); v.feyer@fz-juelich.de (V.F.); 4Fakultät f. Physik and Center for Nanointegration Duisburg-Essen (CENIDE), Universität Duisburg-Essen, D-47048 Duisburg, Germany; 5Istituto Officina dei Materiali (IOM), Consiglio Nazionale delle Ricerche (CNR), Lab. TASC, S.S. 14, Km. 163,5, 34149 Trieste, Italy; floreano@iom.cnr.it

**Keywords:** density functional theory, transition metal porphyrinato complexes, molecular cluster model

## Abstract

Density functional theory, combined with the molecular cluster model, has been used to investigate the surface *trans*-effect induced by the coordination of small molecules L (L = CO, NH_3_, NO, NO_2_ and O_2_) on the cobalt electronic structure of cobalt tetraphenylporphyrinato (CoTPP) surface-supported on coinage metal surfaces (Cu, Ag, and Au). Regardless of whether L has a closed- or an open-shell electronic structure, its coordination to Co takes out the direct interaction between Co and the substrate eventually present. The CO and NH_3_ bonding to CoTPP does not influence the Co local electronic structure, while the NO (NO_2_ and O_2_) coordination induces a Co reduction (oxidation), generating a 3d^8^ Co^I^ (3d^6^ Co^III^) magnetically silent closed-shell species. Theoretical outcomes herein reported demonstrate that simple and computationally inexpensive models can be used not only to rationalize but also to predict the effects of the Co–L bonding on the magnetic behaviour of CoTPP chemisorbed on coinage metals. The same model may be straightforwardly extended to other transition metals or coordinated molecules.

## 1. Introduction

Transition metal porphyrinato (MP) and phthalocyaninato (MPc) complexes hold a prominent position among countless adsorbates because of their unique characteristics. Indeed, their conformational flexibility allows them to adopt different structural arrangements on diverse substrates, which may significantly influence their properties. Moreover, the metal centre occupying either the P^2−^ or the Pc^2−^ four-fold coordinative pocket often plays an active role in catalysis and sensors [1,2,3,4,5,6,7], while the M *n*d*^m^* configuration drives the corresponding magnetic behaviour [8,9,10,11]. All these features make surface-supported MP and MPc appealing for technological applications, including chemical sensors [11,12], storage [13], spintronic/magnetic devices [14,15], and heterogenous catalysis [12,16]. With specific reference to spintronic applications, the ability to finely tune the adsorbate/substrate spin interaction is crucial and demands a detailed understanding of the interphase electronic properties. In this context, recent studies [17] have also shown that a chemical stimulus, such as the occupation of a M coordinative vacancy by a small molecule L, may be exploited to control the interphase magnetic properties by “switching on”/“switching off” the spin interactions between L, M, and the substrate (S) [18,19,20,21,22]. As such, the competition between the S–M and the M–L interactions has frequently been referred to as the surface spin-*trans* effect, or, more simply, as the *trans*-effect [23,24]. Per se, it is noteworthy that, even when dealing with the same MP/MPc surface-supported on the same S, diverse L may generate diverse magnetic behaviours [18,19,20,21,25]. A quantum mechanical modelling of the S–MP–L (S–MPc–L) interphase electronic properties might then be a promising alternative to an inefficient trial-and-error approach for designing new spintronic/magnetic devices [14,15].

In general, numerical experiments carried out by combining the density functional theory (DFT) with the supercell approximation (periodic DFT calculations) provide very accurate results, thus allowing for insights into the molecular, electronic, and magnetic properties of the surface-supported MP and MPc [25]. Nevertheless, a major drawback of such an approach concerns its computational cost, which may become prohibitive when several degrees of freedom are considered, or a defective surface is tackled. Magnetic properties of surface-supported MP and MPc are mainly associated with the M oxidation and spin states [26], i.e., with highly localized properties whose modelling within the molecular cluster approximation is not only computationally less demanding than the ordinary periodic model but also physically reasonable [27]. As a matter of fact, the molecular cluster model allows to look “into the changes in the local bonding situation at the metal centre upon coordination” and “has the advantage that a direct connection to molecular coordination chemistry can be made” [25].

The chemisorption of cobalt tetraphenylporphyrinato (CoTPP) on diverse metallic S including Ni [23] and coinage metals (Cu [26,28,29], Ag [20,25,26,30,31], Au [18,20,21,28,30]) has been investigated in detail in the past. In this regard, it has been shown that both Cu and Ag surfaces act as electron donors, thus inducing a Co^II^ → Co^I^ pseudo reduction [20,25,26,28,30,31], while the Au surface does not affect the Co oxidation state [18,20,21,32]. Even though a rationale for these evidences has been provided by periodic DFT numerical experiments [18,21,25,26,29,33], it has been shown that less expensive calculations may adequately model experimental results (“switching on”/“switching off” the spin interaction) [15,17,25].

In this contribution, we present and discuss the outcomes of a series of DFT-based calculations, carried out by adopting the molecular cluster model to rationalize and hopefully predict the magnetic behaviour of CoTPP surface-supported on different metallic surfaces (Figure 1), herein mimicked by a single atom (hereafter, MS; see below) and interacting at the same time with diverse L (L = CO, NH_3_, NO, NO_2_ and O_2_, Figure 1).

Even though, at first glance, the adopted model might be considered an oversimplification, it is useful to remind that Hieringer et al. [25] theoretically investigated the surface *trans*-effect of MPs (Co, Fe, Zn) on Ag(111) by means of molecular cluster calculations run by attaching either a single Ag atom or a small cluster of varying size (Ag_19_, Ag_72_) to the MP metal centre. The outcomes they obtained by exploiting diverse Ag*_n_* (*n* = 1, 19, 72) clusters as representative of the Ag(111) surface quantitatively differ; nevertheless, all of them perfectly agree about “the strong trans effect of NO on the interaction of CoTPP with the Ag substrate”. The authors also emphasized that the adoption of the Ag_19_ or Ag_72_ clusters, besides been computationally more demanding, has “… the drawback that an interpretation of the results is less straightforward than for the (NO)(Ag)MTPP models” [25].

## 2. Computational Details

Geometrical parameters of the adopted clusters have been optimized without any constraint by exploiting the Amsterdam Density Functional (ADF) suite of programs [34,35]. Scalar relativistic (SR) spin-restricted/spin-unrestricted calculations have been carried out by adopting the Zeroth Order Regular Approximation (ZORA [36,37]), by using the GGA functional BP86 [38,39], by employing a triple-ζ with one polarization function (TZP) Slater-type basis set [40] for all the atoms, and by freezing throughout the calculations the C, N, and O 1s atomic orbital (AO), the Co and Cu 1s–2p AOs, the Ag 1s–3d AOs, and the Au 1s–4d AOs. Incidentally, the same computational set up has been successfully adopted to investigate the ground state properties of a quite large number of M complexes having either Pc^2−^ or TPP^2−^ as ligands [15,17,41,42,43,44,45]. 3D contour plots (CPs) have also been obtained to acquire information about the localization and the character of the frontier MOs. Finally, bonding energies (*BE*s) have been analysed by means of Ziegler’s extended transition state [46] method. According to this scheme, *BE*s may be written as
(1)$$BE=-\left(\Delta E_{\mathrm{es}}+\Delta E_{\mathrm{Pauli}}+\Delta E_{\mathrm{orb}}+\Delta E_{\mathrm{prep}}\right)$$
where, Δ*E*_es_, Δ*E*_Pauli_ and Δ*E*_orb_ represent contributions due to the pure electrostatic interaction, the Pauli repulsion (hereafter Δ*E*_es_ + Δ*E*_Pauli_ = Δ*E*_sr_, the steric repulsion), and the orbital interaction, respectively. The last term Δ*E*_prep_ provides information about the energy required to relax the geometrical structure of the CoTPP and L fragments to the geometry they assume in the final cluster. In this regard, it is noteworthy that atomic fragments from which a molecule/cluster is built must be spin-restricted [35]. Both MS–CoTPP and MS–CoTPP–L *BE*s have been then corrected by $\Delta E_{\mathrm{sp}}^{\mathrm{MS}}$, which corresponds to the −Δ*BE* between a spin-unrestricted and a spin-restricted atomic fragment. Amendments to *BE*s due to the basis set superposition error have been systematically ignored as their contribution is known to be minute [47].

## 3. Results and Discussion

The competition between the S–Co and Co–L interactions, as well as its influence on the CoTPP magnetic properties, has been investigated by adopting the following three models: (i) MS–CoTPP (MS = Cu, Ag, Au), representing the surface-supported CoTPP; (ii) CoTPP–L (L = CO, NH_3_, NO, NO_2_, O_2_), corresponding to the diverse adducts herein considered; and (iii) MS–CoTPP–L, representing the interphase generated by coordinating L to the surface-supported CoTPP. Optimized Cartesian coordinates of the free L molecules, of the free CoTPP complex, and of the MS–CoTPP, CoTPP–L, and MS–CoTPP–L clusters are reported in Appendix A of the Appendix A, while *BE*s and oxidation states for each molecule/adduct/cluster herein considered are collected in Appendix A of the Appendix A.

### 3.1. CoTPP on Cu, Ag and Au Substrates 

The S–CoTPP interaction has been herein modelled by considering the direct interaction of Co with a single Cu, Ag, or Au atom, labelled MS (see Figure 1). Even though perfectly aware that (i) CoTPP on Ag(111) occupies the surface hollow sites [48] and (ii) CoP [49] and CuP [50] on Cu(111) sit on the bridge sites, similarly to Hieringer et al. [25], who tested the *BE* of CoTPP on diverse Ag chemisorption sites finding *BE* minute variations, we decided to choose the on-top site for the sake of simplicity and to take full advantage of symmetry. As such, it is noteworthy that Buimaga-Iarinca and Morari [51] explored in great detail the effect of translation on the *BE* for MP adsorbed on Ag(111), finding a tiny energy dispersion (0.1 eV) among the different adsorption sites.

The neutral Cu, Ag, and Au atoms have a 3d^10^4s^1^, 4d^10^5s^1^, and 5d^10^6s^1^ electronic configuration, respectively, while the low-spin (LS) Co^II^ species occupying the centre of the TPP^2−^ coordinative pocket carries a single unpaired electron in the Co 3$d_{z^{2}}$-based molecular orbital (MO) [52]. Thus, the C_4v_ [53] MS–CoTPP cluster may have either 0 or 2 unpaired electrons. In the absence of any constraint on the MS–Co internuclear distance, the antiferromagnetic coupling between the two unpaired electrons is estimated to be significantly and systematically more stable than the ferromagnetic one by 0.71, 0.87, and 0.56 eV for MS = Cu, Ag, and Au, respectively. Moreover, when relativistic effects are taken into account [54], the MS–Co internuclear distances and the MS–Co *BE*s corresponding to the spin-paired configuration (see Appendix A of the Appendix A) have the well-known trend within the triad, thus indicating that, among the MS–CoTPP interactions, the Ag–CoTPP one is the weakest and most labile. Incidentally, MS–Co *BEs* amount to 1.17, 0.91, and 1.11 eV for MS = Cu, Ag, and Au, respectively (see Appendix A of the Appendix A).

Although of some interest to grasp the main features of the MS–CoTPP interaction, *BE*s and internuclear distances are unable to rationalize the CoTPP “switch on” → “switch off” magnetism upon chemisorption on Cu [26,28,29,33] and Ag [20,25,26,30,31,33], as well as the absence of any CoTPP demagnetization upon chemisorption on Au [18,20,21,32,33]. The thorough analysis of the MS–CoTPP frontier MOs turns out to be a Hobson’s choice to obtain a rationale for the experimental trend. The MS–CoTPP interaction may be roughly described by a two-electrons/two-orbitals model involving the Co 3$d_{z^{2}}$-based singly-occupied MO (SOMO) and the MS (*n* + 1)s AO (*n* = 4, 5, and 6 for Cu, Ag, and Au, respectively). The analysis of their in-phase and out-of-phase combinations (^MS–Co^σ and ^MS–Co^σ* in Figure 1) suggests that, upon chemisorption of CoTPP on Cu and Ag, a Co^II^ + MS^0^ → Co^I^ + MS^I^ pseudo redox reaction takes place (see Figure 1D). Consistently with the presence of a pseudo redox reaction involving a net Cu^0^ → Co^II^/Ag^0^ → Co^II^ charge transfer, the completely occupied ^Cu–Co^σ and ^Ag-Co^σ combinations are mainly localized on the Co 3$d_{z^{2}}$-based MO, while the empty ^Cu–Co^σ^*^ and ^Ag-Co^σ^*^ ones are strongly localized (>50%) on the ^Cu^4s-based and the ^Ag^5s-based MO, respectively. As such, it can be also useful to mention that the Cu and Ag Hirshfeld charges (Q) [55] of the Cu–CoTPP and Ag–CoTPP C_4v_ clusters amount to 0.19 and 0.24, respectively (see Appendix A of the Appendix A). Incidentally, the LS state foreseen by the molecular cluster model herein adopted and implying the presence of a pseudo 3d^8^ Co^I^ species well agrees with DFT periodic calculations modelling the CoTPP chemisorption on Cu and Ag surfaces [20,25,26,28,29,30,31,33].

A change of scenery takes place when MS = Au; besides the comparable localization percentage of the ^Au–Co^σ^*^ MO on the Co 3$d_{z^{2}}$ and Au 6s AOs (37% and 42%, respectively), the Au–CoTPP ^Au^Q is close to zero (0.05). In other words, no Au^0^ → Co^II^ charge transfer able to “switch off” the CoTPP magnetization upon chemisorption on Au seems to be present [18,21,32,33]. In summary, the obtained results based on the molecular cluster model provide a rationale of the magnetic behaviour of CoTPP upon chemisorption on coinage metals. Notably, such an approach is undoubtedly simpler and computationally less expensive than periodic DFT calculations.

### 3.2. CoTPP–L Adducts (L = CO, NH_3_, NO, NO_2_, O_2_)

The L selection has been determined by the presence in the literature of experimental and theoretical data pertaining to the CoTPP–L adducts [24,25,33]. The L herein considered may be divided in two groups according to their diamagnetic (CO and NH_3_, hereafter ^0^L) or paramagnetic (NO and NO_2_, ^1^L; O_2_, ^2^L) nature, where the superscripts 0, 1, and 2 simply refer to the number of unpaired electrons carried by L. The valence manifold of the CoTPP–^0^L adducts is then unavoidably characterized by the presence of a single unpaired electron [24,33] carried by Co^II^, while the scenery may be more multifaceted when CoTPP–*^k^*L adducts (*k* = 1 or 2) are considered.

Both CO and NH_3_ bind CoTPP vertically, C- and N-down oriented [33]; molecular cluster calculations have been then carried out by assuming a C_4v_ and a C_s_ symmetry [53] for the CoTPP–CO and CoTPP–NH_3_ species, respectively (see Appendix A of the Appendix A). CoTPP–CO and CoTPP–NH_3_
*BE*s, estimated according to Equation (1) are 0.33 and 0.28 eV, respectively, and are reported in Appendix A of the Appendix A together with the optimized Co–D bond lengths (BLs, D corresponds to the L donor atom) and the Nalewajski-Mrozek (^NM^I_Co–D_) bond multiplicity indexes [56,57,58,59,60,61,62]. In this regard, it can be useful to mention that the optimized BL_C–O_ passes from 1.139 to 1.157 Å upon moving from the free molecule to the coordinated one, while both the N–H BLs and the H− N ^−H bond angles (BA) of the coordinated NH_3_ are negligibly affected upon coordination.

As expected and anticipated, the electronic structure analysis reveals that the ^0^L coordination to CoTPP does not determine any relevant charge transfer able to modify the Co^II^ oxidation state [24,33]. In addition, the spin population analysis confirms that Co maintains its unpaired electron upon coordination, which remains localized on the Co-based 3d AOs (see Figure 2 and Appendix A of the Appendix A). Even though only two ^0^L have been herein considered, it appears likely that the ^0^L bonding to CoP-like and CoPc-like molecules cannot significantly perturb their magnetic properties.

Similarly to CoTPP–^0^L, the ^1^L coordination to CoTPP has been explored by taking advantage of the available experimental evidence [18]. In more detail, ADF calculations have been run by assuming a N-down orientation for both NO and NO_2_ and by adopting a C_s_ and a C_2v_ symmetry [53] for the CoTPP–NO and CoTPP–NO_2_ adducts, respectively (NO and NO_2_ O atoms point toward the meso C atoms of the macrocycle). Optimized geometries (see Appendix A of the Appendix A) and *BE*s have been evaluated for both LS (no unpaired electron) and high spin (HS, two unpaired electrons) states.

Starting from CoTPP–NO, its diamagnetic state (the LS state is found to be 1.05 eV more stable than the HS state) and the peculiar geometry of the [CoNO]^8^ fragment (the superscript 8 indicates the total number of electrons mostly localized on the Co 3d and the NO π^*^ orbitals) closely resemble the spin state and the crystal structure of the [Co(NO)(Salen)] adduct (Salen = N,N’-bis(salicylidene)ethylenediamine) [63]. More specifically, the Co− N ^−O BA is far from being linear both in CoTPP–NO (122.5°, see Appendix A of the Appendix A) and in [Co(NO)(salen)] (127.0° [63]), the Co–N BLs are nearly identical (1.805 Å in CoTPP–NO, see Appendix A of the Appendix A, and 1.807 Å in [Co(NO)(salen)] [63]), and the Co^II^ species lies significantly above the plane passing through the donor atoms of the four-fold coordinative pocket (0.19 Å in CoTPP–NO and 0.25 Å in [Co(NO)(salen)] [63]). Incidentally, the CoTPP–NO highest occupied MO (the 76a’ HOMO) is strongly localized on the {CoNO}^8^ fragment (66%), and it accounts for a bonding interaction, σ in character (see Figure 2 and Appendix A of the Appendix A), between the ^NO^π^*^_||_ MO and the Co 3d-based AOs lying in the C_s_ symmetry plane, while the 51a” lowest unoccupied MO (LUMO) is reminiscent of the ^NO^π^*^_⊥_ MO (see Appendix A of the Appendix A).

Despite all these similarities, the different behaviour of the {CoNO}^8^ N–O BL upon moving from the free molecule to the coordinated one must be underlined. Theoretical results herein reported show a slight lengthening upon coordination (from 1.166 to 1.185 Å), while the experimental BL of the free NO (1.15 Å) perfectly matches the [Co(NO)(salen)] one [63]. It is also worth noting that, even though the geometry of the {CoNO}^8^ fragment herein optimized fits very well the one obtained by Kim et al. for CoTPP–NO by means of periodic DFT calculations [64], their Co–NO *BE* (1.67 eV) is higher than what we obtained by exploiting the molecular cluster model (1.32 eV, see Appendix A of the Appendix A). This difference could be due to the different exchange-correlation potentials adopted in periodic (PBE [65]) and molecular cluster (BP86 [38,39]) numerical experiments. As a whole, our data indicate that: (i) the CoTPP–NO bonding is accompanied by the Co^II^ → Co^I^ “reduction”; (ii) the saddle conformation adopted by Kim et al. for the CoTPP–NO on Au(111) takes place independently of the surface presence; and (iii) if $\Delta E_{\mathrm{prep}}^{\mathrm{NO}}$ (0.02 eV) and $\Delta E_{\mathrm{prep}}^{\mathrm{CoT}\mathrm{PP}}$ (0.21 eV) are both neglected, the Co–NO *BE* is rather close to the value estimated by Kim et al. for the free CoTPP–NO [64].

Notwithstanding the lack of experimental evidence for the CoTPP–NO_2_ adduct, just under thirty years ago Rousseau et al. [66] tackled the electronic and molecular properties of the EPR silent CoPc–NO_2_ adduct, pointing out that the Co–N_NO__2_ direct interaction characterized by the Y-shaped coordination of NO_2_ to CoPc is accompanied by a Co → NO_2_ charge transfer able to affect the electronic density on the pyrrolic N atoms.

Analogously to CoTPP–NO, the CoTPP–NO_2_ diamagnetic state (LS) is found to be 0.91 eV more stable than the paramagnetic one; moreover, the optimized geometrical parameters of the C_2v_ CoTPP–NO_2_ adduct (see Appendix A of the Appendix A) are very close to those adopted by Rousseau et al. (Co–D = 1.91 Å; Co− N ^−O = 120.0°) [66] for their qualitative extended Hückel numerical experiments. As a result, the NO_2_ geometry changes upon coordination to CoTPP (the optimized N–O BL lengthens from 1.214 to 1.235 Å and the O− N ^−O BA narrows from 133.2° to 125.3°) are perfectly in tune with the above-mentioned Co → NO_2_ charge transfer. Incidentally, the experimental N–O BL and O− N ^−O BA values pass from 1.197 Å and 134.3° in the free NO_2_ to 1.236 Å and 115.4° in the NO_2_^−^ nitrite ion [67].

Considering the electronic properties of CoTPP–NO_2_, a thorough analysis of its frontier MOs (see Figure 2) reveals that, contrary to the Co–NO bonding, the Co–NO_2_ one is accompanied by the Co^II^ → Co^III^ oxidation. In more detail, the CoTPP–NO_2_ 45a_1_ HOMO is reminiscent of the ^NO^^2^π^*^_||_ SOMO [68] and it is poorly localized (9%) on the Co 3d-based AOs (see Figure 2D), while the 46a_1_ LUMO is strongly concentrated (46%) on the Co $3d_{z^{2}}$ AO. As a whole, even though the coordination to CoTPP of both the ^1^L herein considered “switches off” the magnetization of the complex (no unpaired electron is present), the ^1^L quenching mechanism is opposite in NO and NO_2_: in the former case, it implies the reduction of the Co^II^ centre through the redox reaction NO + Co^II^ → NO^+^ + Co^I^, while in the latter, the oxidation of the Co^II^ centre through the redox reaction NO_2_ + Co^II^ → NO_2_^−^ + Co^III^ takes place.

M complexes with O_2_ as a ligand have attracted a great interest, mainly driven by the need of looking into the nature and strength of the bonding between dioxygen and M in proteins involved in the O_2_ carriage in living things. According to their non-linear M–(η^1^-O–O) or bridging M–(μ-O_2_) configuration, the two main types of 1:1 M–O_2_ complexes are usually labelled superoxo and peroxo [67]. Superoxo Co complexes are well structurally characterized [69,70,71,72,73,74,75] and Co–(η^1^-O_2_) and O–O BLs (Co – O ^ –O BA) range between 1.85–1.90 and 1.25–1.35 Å, respectively (115°–125°).

Numerical experiments herein reported have been limited to the superoxo configuration [18] for which both the HS (three unpaired electrons) and LS (one unpaired electron) states have been explored. Analogously to CoTPP–^1^L, the LS state appears more stable than the HS one (0.49 eV); moreover, the optimized Co–(η^1^-O_2_) BL and Co – O ^ –O BA (1.858 Å and 119.7°, see Appendix A of the Appendix A) perfectly fall in the above reported ranges (the optimized O–O BLs in the free molecule and in the (η^1^-O_2_) species are 1.235 and 1.286 Å, respectively). A thorough analysis of the CoTPP–(η^1^-O_2_) electronic structure reveals that, similarly to CoTPP–NO_2_, the Co–(η^1^-O_2_) bonding is accompanied by the Co^II^ → Co^III^ oxidation with all but one spin orbitals reminiscent of the O_2_ π_g_ MOs [67] (antibonding with respect to the O–O interaction) occupied; the fourth, π_g_-like unoccupied MO (VMO) corresponds to the CoTPP–(η^1^-O_2_) 51a” LUMO, completely localized (86%) on the spin down (↓) component of the π_g_ spin orbital ⊥ to the symmetry plane (see Figure 2E). Consistently with such a picture, the Co spin density is negligible, and the single unpaired electron is completely localized on O_2_ (see Appendix A of the Appendix A). Interestingly, even though the number of unpaired electrons does not vary upon moving from CoTPP to CoTPP–(η^1^-O_2_) and no magnetization “switch off” is then expected (see Figure 2F), the spin configuration of the superoxo adduct is completely different from that of the pristine complex [33]. A summary of the oxidation state of the Co and L after the coordination is reported in Appendix A of the SM.

### 3.3. CoTPP–L Adducts on Cu, Ag and Au Substrates: The Trans-Effect

Theoretical results pertinent to the MS–CoTPP clusters and the CoTPP–L adducts confirm that simple, tiny, and computationally inexpensive models may be adopted to acquire information about the magnetic behaviour of CoTPP upon chemisorption on coinage metals as well as on perturbations induced by the coordination of ^*k*^L (*k* = 0, 1, and 2) on the CoTPP frontier orbitals. The feasibility testing of the same approach to explore the effects induced at the same time by chemisorption and coordination on the CoTPP electronic structure is then challenging on one hand and appealing in terms of computational costs on the other hand. As a result, it may be useful to remember that, while both experimental and theoretical data are available in the literature, the latter studies have mostly been carried out by adopting periodic calculations [18,19,21,23,24,25,33,76]. Nature, symmetry, and strength of the surface *trans*-effect characterizing the different S–CoTPP–L interphases have been herein investigated by adopting the MS–CoTPP–L clusters, representative of the L interaction with CoTPP deposited on S. MS–CoTPP–L theoretical outcomes have been then compared with experimental and/or theoretical data from the literature, when available.

MS–CoTPP–^0^L results have several common features for both the ^0^L herein considered, the most relevant being: (i) the MS–CoTPP–CO LS state (no unpaired electron) is more stable than the HS one (two unpaired electrons) by 0.58, 0.52, and 0.70 eV for MS = Cu, Ag and Au, respectively; (ii) similarly, the MS–CoTPP–NH_3_ LS state is more stable than the HS one by 0.77, 0.61, and 0.82 eV for MS = Cu, Ag and Au, respectively; iii) both the MS–Co and the Co–D *BEs* decrease upon moving from MS–CoTPP and CoTPP–^0^L to MS–CoTPP–^0^L for Cu and Ag (Appendix A of the Appendix A); iv) both the MS–Co and the Co–D BLs increase upon moving from MS–CoTPP and CoTPP–^0^L to MS–CoTPP–^0^L for Cu and Ag (Appendix A of the Appendix A); and v) the MS–Co interaction weakening induced by the ^0^L coordination is accompanied by the MS^I^ + Co^I^ → MS^0^ + Co^II^ redox reaction for MS = Cu and Ag. Moreover, the MS–CoTPP–^0^L number of unpaired electrons (disregarding that localized on MS) mirrors the CoTPP–^0^L one. Incidentally, the doubly occupied MS–CoTPP–CO 30a_1_ HOMO (MS = Cu and Ag) is strongly localized on the MS (*n* + 1)s/*n*$d_{z^{2}}$ (*n* = 3 and 4 for MS = Cu and Ag, respectively) AOs and the Co 3$d_{z^{2}}$ AO, while the doubly occupied MS–CoTPP–NH_3_ 74a′ HOMO (MS = Cu and Ag) is strongly concentrated on the MS (*n* + 1)s/*n*$d_{z^{2}}$/*n*$d_{x^{2}-y^{2}}$ (*n* = 3 and 4 for MS = Cu and Ag, respectively) AOs and the Co 3$d_{z^{2}}$/3$d_{x^{2}-y^{2}}$ AOs. Incidentally, MS–CoTPP–CO theoretical outcomes perfectly agree with experimental [76] and theoretical [33] data in the literature.

Even though Au–CoTPP–^0^L ADF results are also consistent with the presence of Co^II^ (3d^7^) and Au^0^ (5d^10^6s^1^) species, it must be remarked that no Co oxidation state variation takes place upon moving from Au–CoTPP to Au–CoTPP–^0^L: a consequence of the absence of any Co^II^ → Co^I^ reduction accompanying the chemisorption of CoTPP on Au. As such, it is noteworthy that the lengthening of the Au–Co and Co–D BLs upon the L coordination is less significant than that estimated for Cu–CoTPP–^0^L and Ag–CoTPP–^0^L (see Appendix A). Molecular cluster results pertinent to Au–CoTPP–NH_3_ agree very well with experimental data and periodic calculations [18,21].

As a whole, the magnetization of the MS–CoTPP interphase (MS = Cu and Ag) is “switched on” by the ^0^L chemisorption, while no variation is expected on passing from Au–CoTPP to Au–CoTPP–^0^L (see Figure 3, third and fourth columns).

Theoretical results obtained for the adducts CoTPP–^1^L and CoTPP–^2^L induce us to foresee that the electron exchange taking place at the S–CoTPP–*^k^*L (*k* = 1 and 2) interphases should not be limited to CoTPP and S, possibly involving ^1^L and ^2^L too. The inspection of Appendix A of the Appendix A confirms this expectation and clearly shows that the strongest MS–CoTPP–*^k^*L *trans*-effect is associated to the NO coordination, whose presence: (i) decreases the *BE*_MS–Co_ (1.17 → 0.41 eV, 0.91 → 0.26 eV, 1.11 → 0.49 eV, for Cu, Ag and Au, respectively); (ii) increases the BL_MS–Co_ (2.27 → 2.36 Å, 2.47 → 2.60 Å, 2.46 → 2.56 Å, for Cu, Ag and Au, respectively). Further insights into the MS–Co and the Co–NO interactions may be gained by referring to Figure 4, where MS–CoTPP–*^k^*L (*k* = 1, 2) simplified energy level diagrams are displayed together with 3D plots of the MOs mainly localized on the MS-based *n*s AO (*n* = 4, 5 and 6 for Cu, Ag and Au, respectively), the Co-based 3d AOs, and the *^k^*L-based π^*^ fragment MOs.

The MO localization percentages reported in the figure testifies to the negligible perturbation induced by the MS presence on the Co–NO bond; moreover, it is noteworthy that the MS (*n* + 1)s-based 83a’ SOMO (MS = Cu, *n* = 3; MS = Ag, *n* = 4) has a negligible contribution (≤2%) from the Co 3d-based AOs. In addition, both the ^NO^π^*^_⊥_-based ^↓^^/^^↑^54a” and the ^NO^π^*^_||_-based ^↑/↓^85a′ MOs have a VMO character, and all but one (the ^↓^^/^^↑^56a” MO) Co 3d-based MOs are completely occupied. The NO *trans*-effect is then characterized by the transfer of the ^1^L unpaired electron to the MS–CoTPP system, prompting Co and MS to assume a 3d^8^ (Co^I^) and a *n*d^10^(*n*+1)s^1^ (Cu^0^/Ag^0^) electronic configuration and taking out, in agreement with periodic calculations [33], any direct MS–Co interaction. Altogether, theoretical outcomes clearly indicate that, even though both the Co oxidation state and its electronic configuration are nearly identical in MS–CoTPP, CoTPP–NO and MS–CoTPP–NO, the NO coordination to MS–CoTPP reduces MS^I^ to its elemental oxidation state through the NO → NO^+^ oxidation inhibiting at the same time any direct MS–Co bonding (see Appendix A). The NO coordination does not induce any “switch on” effect on MS–CoTPP because the Co species does not vary its oxidation state and the ^1^L unpaired electron is used to reduce MS to its elemental oxidation state. It is noteworthy that both experimental studies and period calculations on S–CoTPP–NO support the molecular cluster outcomes herein reported [19,76].

The comparison of the Au–CoTPP–NO frontier electronic structure with those of the Cu–CoTPP–NO and Ag–CoTPP–NO molecular clusters clearly indicates that the different behavior of Au compared to Cu and Ag has, ultimately, to be traced back to the diverse MS–CoTPP interaction on passing from Cu/Ag to Au (look at the second column of Figure 3). Au preserves its 5d^10^6s^1^ electronic configuration along the whole Au → Au–CoTPP → Au–CoTPP–NO path, while the NO → NO^+^ oxidation (both the ^NO^π^*^_⊥_-based and the ^NO^π^*^_||_-based Au–CoTPP–NO MOs are empty) provides the electron needed for the Co^II^ → Co^I^ reduction. As a whole, the NO coordination to CoTPP chemisorbed on Au is expected to have a “switch off” effect due to the generation of a Co^I^ 3d^8^ closed shell (see Figure 3).

Although not as strong as that induced by the NO coordination, the NO_2_
*trans*-effect is quite effective too (see Appendix A of the Appendix A); in particular, upon moving from MS–CoTPP and CoTPP–NO_2_ to MS–CoTPP–NO_2__,_ the *BE*_MS–Co_ and *BE*_Co–D_ decreasing (see Appendix A) is accompanied by the BL_MS–Co_ and BL_Co–D_ increasing (see Appendix A). Common features of the MS–CoTPP–NO_2_ bonding scheme are: (i) the strong localization of the SOMO on the MS (*n* + 1)s AO (see Figure 4), which is consistent with a MS *n*d^10^(*n* + 1)s^1^ elemental electronic configuration (*n* = 3, 4, 5 for MS = Cu, Ag, Au, respectively); (ii) the NO_2_ “closed-shell” nature after coordination and then its “nitrite” character (see above); and (iii) the local “closed shell” electronic configuration (3d^6^) of Co oxidized by NO_2_ to Co^III^ (only the six t_2g_-like Co-based spin orbitals {the ^↑/↓^35b_1_, ^↑/↓^33b_2_, ^↑/↓^46a_1_ levels in Cu–CoTPP–NO_2_; the ^↑/↓^35b_1_, ^↑/↓^33b_2_, ^↑/↓^48a_1_ levels in Ag–CoTPP–NO_2_; the ^↑/↓^35b_1_, ^↑/↓^37b_2_, ^↑/↓^50a_1_ levels in Au–CoTPP–NO_2_} are occupied). Even though the MS–CoTPP–NO_2_ magnetic behaviour is closely reminiscent of the CoTPP–NO_2_ one (no “switch on” effect takes place upon chemisorption of NO_2_), it must be underlined that, similarly to MS–CoTPP–NO, the Au–CoTPP–NO_2_ frontier electronic structure differs from the Cu–CoTPP–NO_2_ and Ag–CoTPP–NO_2_ ones as a consequence of the diverse MS–CoTPP interaction on passing from Cu/Ag to Au. The NO_2_ chemisorption on the Au–CoTPP species has then a “switch off” effect on the CoTPP magnetization. Once again, these results agree with experimental data and periodic calculations on Au-CoTPP-NO_2_ [18,21].

Any attempt to optimize the geometry of the MS–CoTPP–(η^1^-O_2_) cluster with the LS configuration (no unpaired electrons) failed, while the HS (two unpaired electrons) geometry optimization converged rapidly and smoothly along the MS triad. Such a peculiar behaviour has to be traced back to the electron transfer processes involving at the same time the Co, MS, and O_2_ species (see below).

Aside from the expected decreases in *BE*_MS–Co_ and *BE*_Co–O_, and the BL_MS–Co_ and BL_Co–O_ increasing upon moving from MS–CoTPP and CoTPP–(η^1^-O_2_) to MS–CoTPP–(η^1^-O_2_) (see Appendix A), a thorough analysis of the MS–CoTPP–(η^1^-O_2_) frontier electronic structure revealed: (i) the strong localization of the SOMO on the MS (*n*+1)s AO (see Figure 4), which is consistent with a MS *n*d^10^(*n*+1)s^1^ elemental electronic configuration (*n* = 3, 4, 5 for MS = Cu, Ag, Au, respectively); (ii) the occupied character of all but one (the ^↓^54a“ level in Cu–CoTPP–(η^1^-O_2_) and Ag–CoTPP–(η^1^-O_2_); the ^↓^57a” level in Au–CoTPP–(η^1^-O_2_)) π_g_ O_2_-based spin-orbitals (the ^↑^50a“, ^↑^77a′, ^↓^81a′, ^↓^54a“ levels in Cu–CoTPP–(η^1^-O_2_); the ^↑^52a“, ^↑^79a′, ^↓^81a′, ^↓^54a“ levels in Ag–CoTPP–(η^1^-O_2_); the ^↑^55a”, ^↑^83a’, ^↓^85a’, ^↓^57a“ in Au–CoTPP–(η^1^-O_2_)), thus indicating the superoxide nature of the coordinated O_2_; and (iii) the occupied character of only six (the ^↓^53a“, ^↑^54 a“, ^↓^77a′, ^↑^78a′, ^↑/↓^82a′ levels in Cu–CoTPP–(η^1^-O_2_), the ^↓^53a“, ^↑^54 a“, ^↓^79a′, ^↑^80a′, ^↑/↓^82a′ levels in Ag–CoTPP–(η^1^-O_2_), the ^↓^56a”, ^↑^57a”, ^↓^83a’, ^↑^84a′, ^↑/↓^86a′ levels in Au–CoTPP–(η^1^-O_2_)) Co 3d-based spin-orbitals, thus awarding a local “closed shell” electronic configuration (3d^6^) to Co, oxidized to Co^III^, through the O_2_ → O_2_^−^ reduction.

Even though the O_2_ coordination to CoTPP adsorbed on coinage metals should be accompanied by a magnetization “switch on” effect, it is noteworthy that unpaired electrons are localized on L rather than on Co (see Appendix A of the Appendix A). A summary of the oxidation state of the Co, L, and MS after the coordination is reported in Appendix A.

## 4. Conclusions

A series of shared features binds the coordination of *^k^*L to CoTPP molecule supported on coinage metal surfaces: (i) both *BE*_MS–Co_ and *BE*_Co–D_ (BL_MS–Co_ and BL_Co–D_) decrease (increase) upon moving from MS–CoTPP and CoTPP–L to MS–CoTPP–L; (ii) the MS oxidation state in the MS–CoTPP–L cluster is systematically found equal to 0; (iii) the local electronic structure of the CoTPP–L fragment in the MS–CoTPP–L cluster is very similar to that of the CoTPP–L adducts; and (iv) the different MS–CoTPP–L *BE*_Co–D_ values are scarcely affected by MS, thus confirming the leading role played by the *trans*-coordinate ligand in the weakening of the direct MS–Co bonding. As a whole, the results presented and discussed herein demonstrate that small and computationally inexpensive molecular clusters can be used to confidently predict the influence of different ligands on the surface chemical bonds of adsorbed metalloporphyrins on diverse coinage metals and then be exploited to drive experiments towards the desired outcomes.

## Data Availability

All data have been illustrated in the manuscript and in the Appendix A.

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
