# Peer review of "The Magnetic Behaviour of CoTPP Supported on Coinage Metal Surfaces in the Presence of Small Molecules: A Molecular Cluster Study of the Surface trans-Effect"

_nanomaterials, 2022, doi:10.3390/nano12020218_

Round 1
Reviewer 1 Report
S. Carlotto have submitted an study about the surface trans-effect infuced on several "metal surfaces" by using DFT methdos and molecular model cluster.
This approach could have two main advantadges respect to calculations done with periodic boundary conditions (PBC): i) reduced computational cost and ii) better description on the system from a chemistry point of view focussing on the bond between "metal surface" and CoTPP.
In my opinion, this paper could be published. However, several question must be previously addressed.
1. The authours should justify the selectection of BP86 functional, as wells as basis sets. Could be more adequate to use basis set like Pople ones (for C, H, N, O), or those developed by Dunning?
2. I guess that in other papers PBC are done with programs such as VASP, SIESTA or similar, where available funciontals are more limited. Commonly, PBE pure functional is used. At least, for structural optimization. Would be apropiate to use PBE in this work aitmed at obtaining results to influenced by the functional?
3.One of the main advantadjes of this paper is the reduced computational cost. Have the authors really checked this reduced computational effort?? In my opinion, at least one system (only one metal surface and one only L molecule) should be calculated with PBC as published in the literature to stablish a reference. Thus, results such as computational time as well as those one related with the systems can be assesed.
4. The authors mentioned "metal surfaces". In this paper, there are not metal surfaces. There is only one atom. Thus, i don't know if the use of "surface" is adequate. In this sense, Scheme 1 displays that CoTPP is bonded to "metal surface" througn a bond between the Co and one metal atom. is this proven? Thus, the authors decided to use reduced molecular model. This model doest not describir the possiblity that bonds between metal-surface atoms can be affected by (or they can affect) MS-Co and L-Co bonds. Thus, how is this effect described? I guess that the authours should carried on some calculations where metal surface is described by small metal clusters. For example. 3 atoms in line, a small surface with 9 atoms or related..... This will allow to validate the use only one atom to describe the metal surface.
Author Response
see the attached file Reply Ref. 1

Reviewer 2 Report
The manuscript entitled “The Magnetic Behaviour of CoTPP Supported on Coinage Metal Surfaces in the Presence of Small Molecules: A Molecular Cluster Study of the Surface Trans-Effect” by S. Carlotto and coworkers tackles the problem of spin states in a coordination complex of cobalt, tetraphenylporphyrin and small ligands. The study, although purely computational, is of interest to the nanotechnological community as it can affect our understanding of molecular switching and signalling via chemically induced modification of spin states.
I find the research very competently designed, and described with great detail. The amount of information is large, therefore some parts of the manuscript are tedious to read (e.g. detailed orbital listings on page 11), however it is difficult to find a remedy. The Supplementary Material is also prepared with great detail and I find it very informative. The manuscript is edited with care, and I find no serious typos or omissions worth noting (the only small formatting error is that references 71 and 72 do not fully conform to the journal rules).
The only minor issue I would like the authors to consider is an explicit inclusion of explanation why a single atom of coinage metal is thought sufficient to mimic the surface. The authors point to their earlier work (lines 74-82), but I would like to see it mentioned explicitly whether some studies were performed to check the amount of the electron donation in growing clusters. I do not suspect that the assumed model might be fundamentally wrong, but maybe some effects would be better visible with several Ag/Au atoms, not only one? Please discuss shortly.
Summarizing, I find the manuscript very well written, the research interesting and fitting the scope of the manuscript, therefore I recommend only a minor revision explaining the issue of the cluster size validity.
End of reviewer remarks
Author Response
See the attached file Reply Ref. 2

Reviewer 3 Report
Review attached.

Author Response
See the attached file Reply Ref. 3

Round 2
Reviewer 1 Report
The authors have justified as well as improved those question related with the theoretical level uses and reduced computational time respect to PBC (questions 1-3) .
However, in my opinion, the influence of metal-metal surface bonds on L-CoTTP bond could be not correctly described with this reduced model. I think that some test calculations should be carried out, where metal surface is described throught small atomic cluster
Author Response
see the attached file "New reply to Ref. 1"

Reviewer 3 Report
The authors have addressed all concerns of this reviewer. Hence the paper can be accepted.
Author Response
The Reviewer explicity said: "The authors have addressed all concerns of this reviewer. Hence the paper can be accepted."
I do not think there is anything to add.